# A Simulation-Based Approach for Evaluating the Effectiveness of Robotic Automation Systems in HMR Product Loading

**DOI:** 10.3390/foods13193121

**Published:** 2024-09-30

**Authors:** Seunghoon Baek, Seung Eel Oh, Seung Hyun Lee, Ki Hyun Kwon

**Affiliations:** 1Digital Factory Project Group, Korea Food Research Institute, Wanju 55365, Republic of Korea; whitesh917@gmail.com; 2Department of Biosystems Machinery Engineering, Chungnam National University, Daejeon 34134, Republic of Korea; 3Food Safety and Distribution Research Group, Korea Food Research Institute, Wanju 55365, Republic of Korea; dr51@kfri.re.kr

**Keywords:** food factory, robot simulation, plant simulation, automation, KPIs

## Abstract

The food industry has tried to enhance production processes in response to the increasing demand for safe, high-quality Home Meal Replacement (HMR) products. While robotic automation systems are recognized for their potential to improve efficiency, their high costs and risks make them less accessible to small and medium-sized enterprises (SMEs). This study presents a simulation-based approach to evaluating the feasibility and impact of robotic automation on HMR production, focusing on two distinct production cases. By modeling large-scale and order-based production cases using simulation software, the study identified key bottlenecks, worker utilization, and throughput improvements. It demonstrated that robotic automation increased throughput by 31.2% in large-scale production (Case A) and 12.0% in order-based production (Case B). The actual implementation showed results that closely matched the simulation, validating the approach. Moreover, the study confirmed that a single worker could operate the robotic system effectively, highlighting the practicality of robotics for SMEs. This research provides critical insights into integrating robotics to enhance productivity, reduce labor dependency, and facilitate digital transformation in food manufacturing.

## 1. Introduction

Research on developing automated processes to enhance productivity and reduce labor costs has become a fundamental focus across various industries. Especially the food industry faces unique challenges as it adapts to rapidly changing market trends and evolving consumer demands. It has utilized various methods to process food materials into products to satisfy consumer needs. It has also focused on improving processes such as raw material management, processing, distribution, and quality [1]. Home Meal Replacement (HMR) products have emerged as representative solutions that meet these requirements. HMR refers to prepared foods that can quickly and conveniently replace home-cooked meals and are both easy to store and consume [2]. The increase in single-person households has led to a rise in the consumption of convenience foods, with a focus on ease of use and reduced meal preparation time [3,4]. Shelf-stable HMR products are often produced through a retorting process, a critical step in ensuring food safety and quality by achieving commercial sterilization [5]. In the Korean HMR market, “bibimbap” contained with retorted vegetables was introduced, which has been very portable and easy to prepare [6]. Stew-type HMR products based on meat (ham, sausages), vegetables, and kimchi have also gained popularity. These products, rich in water and various nutrients, are prone to bacterial contamination, necessitating commercial sterilization for shelf stability [7]. This process, typically applied to products sealed in pouches made from bonded single-layer plastic films, metal foils, or laminates, not only extends shelf life, but also requires compliance with stringent safety standards, essential in large-scale automated production systems [8]. Food manufacturing companies are placing more emphasis on sustainability and efficiency in production, while simultaneously maintaining quality and safety standards. Companies are transitioning from traditional manual production processes to the adoption of automated systems, driving digitalization [9].

Robotics have emerged as a critical solution for improving efficiency, safety, and product quality in food manufacturing. Robotic technology, which reduces costs and improves quality, is gaining significant attention [10]. The purpose of robotics applications is to automate dangerous, repetitive, and labor-intensive tasks. Robots can be categorized based on their degrees of freedom and the roles of their joints. Cartesian robots move in straight lines, while six-axis articulated robots are more flexible [11]. Articulated robots are structured like a human arm, providing high flexibility within a confined workspace. This flexibility allows for the design of trajectories suited to specific purposes. This made robots increasingly used alongside existing equipment in various manufacturing industries [12]. Early applications of robotic automation in the food industry were primarily focused on palletizing tasks, which were well-suited for simple, repetitive transport and stacking processes [13]. As robotics technology advanced, research was conducted to select appropriate robots based on food types, conditions, and states [14]. To demonstrate the feasibility of these selected robots performing more precise operations, many studies were conducted in food processing industries [15]. The application of robotics is being studied in the meat industry, particularly in sectors like beef and poultry [16,17,18]. Additionally, research is being conducted to improve processes in the seafood industry [19], like improving salmon filet cutting operations using a 3D vision system combined with robotic automation [20]. Despite such advances, the International Federation of Robotics (IFR) noted that the food and beverage sector showed the lowest application of industrial robots compared to other industries. This is due to the complexity of standardizing robotic applications in the diverse processes of the food sector, compatibility issues among platforms, and limited sharing of integration methods by suppliers. The majority of SMEs remain hesitant due to financial costs and perceived risks, compounded by specific needs for hygiene, worker safety, productivity, and ease of operation.

To solve these problems, simulations have been employed to assess the economic benefits and effectiveness of applying robots before actual implementation [21,22]. Considering that the food industry is a complex and dynamic sector that involves various processes, simulation models play an important role. Simulation has been used as a key decision-making tool. Discrete Event Simulation (DES) is one of the most widely used simulation techniques in manufacturing process research and analysis [23]. It is used to identify bottlenecks in production processes and logistics systems, as well as to compare and analyze results after optimizing production conditions [24]. Previously, block diagram-based logic and scripts were used to create models similar to flowcharts, which represented production quantities, processing times, and other metrics [25]. A small-sized pizza manufacturer analyzed the manufacturing time of each process through simulation and suggested improvement measures to report the possibility of increasing production and efficiency [26]. In addition, a case study in a factory manufacturing frozen French fries and chips emphasized the importance of improving using a simulation. In this factory, it was analyzed that performance was significantly improved compared to the existing production method by optimizing the configuration, including three packaging lines and two cutting process lines [27].

With advancements in technology, factory simulations now allow process flows to be visualized in a virtual space [28,29]. This enables tracking and pre-monitoring of process changes during operations. Simulation models can effectively manage and optimize food production processes by alleviating bottlenecks through simulation analyses of factors such as labor efficiency and material flow [27]. Moreover, Digital Twin (DT) models are increasingly recognized within Industry 4.0, although those involve complex integration of real-world data into digital representations [30]. However, in contrast to individual predictions during production or distribution, food manufacturing companies often are subjected to small quantity batch production. This characteristic, combined with the complexity of production planning, inventory management, and uncertainties, presents significant challenges for SMEs in adopting DT technologies [31]. Despite data limitations and fixed processes that restrict production scheduling in the food industry, some studies have demonstrated the feasibility of DT for efficient operations, such as monitoring and controlling production and supply plans in ice cream factories [32]. While there has been extensive research on using simulations and robotics, there is still a lack of studies demonstrating their actual effectiveness in real-world food industry applications.

The main purpose of this study is to provide a practical case study solution by applying articulated robots in two different manually operated retort product loading processes within automated production lines. Specifically, this research identified a gap between the simulation and real application of HMR manufacturing systems by analyzing key performance indicators (KPIs). The simulations were used to identify existing production challenges, such as worker overload, and to evaluate the effectiveness of robotic automation as a solution. By implementing simulation-based models, the study aimed to predict potential productivity gains and validate these predictions against real application results. The simulation demonstrated the potential to assess increases in production output and compare them with actual outcomes upon implementing automation. To support this study, Discrete Event Simulation (DES)-based Tecnomatix Process Simulation and Plant Simulation were utilized, focusing on increasing line productivity and addressing digitalization challenges faced by SMEs.

## 2. Materials and Methods

Two cases with different scales of retort product production processes were chosen for this study. Using simulation software, this study analyzes the workload of workers in the existing manual production environment and assesses the effectiveness of implementing a robotic automation system. The following methods were employed to compare the performance between the manual and automated processes.

### 2.1. Methodology Description

Plant simulation is an important tool for modeling and optimizing manufacturing processes. Companies use simulation models as a key decision-making tool when determining whether to add a new production line or to improve an existing process. Specific procedures and approaches were followed to achieve the objectives of designing and testing the simulation [33,34]. The process improvement began with gathering data from the current site. Based on the initial objectives, data on production output, target products, and the number of workers were collected. This information was then applied to modeling and simulation to identify bottlenecks in the current process. The improvement simulations were subsequently used to evaluate whether the initial objectives were met through an iterative feedback process. Once the objectives were validated, the simulation results were implemented in actual production environments.

Plant simulation processes were used in this article to achieve the research objective of improving existing manual processes by applying a robotic automation system. Since the accurate analysis of simulation results significantly impacts the success of simulation studies, SIEMENS Tecnomatix Process Simulate (V16.0.1, SIEMENS Inc., Munich, Germany) was utilized. This tool was used to configure robot motions and trajectories that can automatically load products, either assisting or replacing human workers. SIEMENS Tecnomatix Plant Simulation (V2201, SIEMENS Inc., Munich, Germany) was additionally employed to compare the outcomes of the current manual process with the improved robotic automation system.

### 2.2. Designated HMR Process for Case Study

In general, the manufacturing process for retort food products consists of several sequential steps: measuring ingredients, filling, sealing, sterilizing (retorting), cooling, inspecting, and packaging, depending on the type of product [8]. This study focuses on improving repetitive and commonly required processes within the retort manufacturing workflow by applying a robotic automation system. The selected process involves loading products onto trays for sterilization after they pass through metal detection, as shown in Figure 1.

The selected process is critical for maintaining the continuity of the workflow, depending on the production speed of the supplied products. Two cases with different scales of retort product production were chosen. For each case, modeling and simulation were utilized to study methods for automating the production line that loads products onto empty trays. A virtual production facility was recreated to replicate the actual work environment, allowing for a comparative analysis between the current manual production method and the robot-automated system. Factors considered include the current facility design, equipment (such as trolleys and trays), and the workspace between the worker and collaborative robots.

Case A represented a large-scale manufacturing site. In this setup, a turntable was used as an intermediate waiting process to control the flow of production. This setup allowed workers to keep up with the production speed. The process was carried out in a harsh environment where multiple workers were involved. They faced hazardous conditions such as a slippery floor and congested work paths, which increased the risk of accidents.

Case B is a production facility using an order-based production system. In this setup, the production quantity was managed by adjusting the number of workers in the process according to the production schedule and required output. Due to the minimal number of workers involved, there was a higher risk of musculoskeletal hazards for the workers. The process required moving heavy trays of products from one space to another, as the loading area and the retort area were in different locations. This created a cumbersome process where products had to be transported using trolleys.

### 2.3. Application of Robot and End-Effector in Process Simulation

A collaborative robot was selected to improve the process. Commonly known as cobots, collaborative robots are designed to work alongside human operators without posing safety risks [35]. These robots are typically constructed with a lightweight frame and equipped with collision detection control system, making them suitable for shared workspaces. They also operate at a relatively slow speed, prioritizing safety by minimizing risks to human operators and shared workspace objects. An articulated robot with a payload capacity of 20 kg, a reach of 1700 mm, and repeatability of 0.1 mm (Doosan Robotics, Suwon, Republic of Korea, Model: H2017) was utilized. This robot is capable of handling trays with a maximum length of approximately 960 mm and transporting six products at a time, each weighing approximately 3 to 4 kg. The robot model was integrated into the simulation library, which facilitated accurate replication of its movements within the simulated environment. The end-effector used was equipped with 30 vacuum gripping systems designed to load products onto the tray one row at a time. The vacuum pads were supplied by VMECA (Magic Gripper, VMECA Co., Ltd., Incheon, Republic of Korea) and featured Micro 2-stage cartridge dual vacuum cartridges (MC10D) with 2-fold top bellows and an integrated 200-mesh vacuum filter. The technical specifications of the vacuum system include a maximum vacuum pressure of −83 kPa, a maximum suction flow rate of 23 NL/min, and an air consumption rate of 20.3 NL/min at 2.2 bar. This configuration was chosen to ensure secure handling and efficient placement of products during the loading process.

To attach the end-effector to the robot’s manipulator, the tool and frame were configured accordingly in the process simulation, as shown in Figure 2. This setup is essential for defining the tool’s position and orientation relative to the robot, allowing the end-effector to be designated as the Tool Center Point (TCP) for accurate trajectory and motion design. Robots determine spatial constraints based on the TCP and the robot’s body. If the TCP is defined using poses, the robot flange position must be expressed as a pose (position and orientation) relative to the robot base or a reference frame. When programmed to move along a specific path, the TCP follows the actual path. This setup facilitates trajectory planning in Tecnomatix Process Simulate, enabling precise placement of objects, as illustrated in Figure 2.

After completing the initial setup, the robot’s process trajectory and speed were configured. Using Tecnomatix Process Simulate software (V16.0.1), the robot’s initial position, known as the ‘Home Position’, was first selected. This is usually set to an appropriate location for starting the task and the same point to which the robot returns after completing its operations. The robot’s movements were planned by identifying pick-up points for the products and placing positions on the tray, connecting these points using point-to-point (PTP) movement. These generated paths were further refined using various features of the software, including collision detection that allowed the robot’s movements to be tested against the actual layout of the production site. The software detected potential collisions with objects such as conveyors, trays, and other equipment, ensuring safe operation.

In conventional settings, the speed at which workers load products is inconsistent. By adjusting the speed of robots in the simulation-based paths, as shown in Figure 3, it was possible to minimize process time and establish a consistent production rate. Considering worker safety, the robot speed was set at 70% of its maximum capacity. Safety concerns were prioritized; therefore, the robot was programmed to operate at maximum speed only within a specific, short path segment—specifically when placing the gripper at the designated spot and lifting approximately 3 cm upwards—to reduce operational time. Through robot simulation, the feasibility of product loading was assessed in advance by accounting for on-site conditions and tray size (960 mm), enabling the selection of the most suitable robot for the process. By configuring the robot’s movements and adjusting its speed, automated production rates could be achieved faster than those in existing manual settings. While a single quick movement might save only about 0.2 s, the cumulative effect over weekly and monthly production could result in significant time savings.

### 2.4. Chosen KPIs

The objective of both cases was to enhance the production process in order to increase throughput per hour or, at the very least, maintain productivity through the implementation of a robot-automated system. Improvement could be achieved in several ways, with the current bottlenecks and product imbalance representing the main obstacles to overcome. To evaluate the impact of the enhanced simulation on these factors, a series of key performance indicators (KPIs) were selected for analysis. Only production-related KPIs were used for evaluation. The selected KPIs were throughput, utilization rate, and human resources. The modeling and logical sequence were developed using the standard library tools of Tecnomatix Plant Simulation from Siemens (V2201). Each of these basic KPIs reflected performance aspects derived from simulation based on monitored and measured data from the actual site. The KPIs could be grouped into similar categories to reduce complexity [36].

In the existing process, throughput was derived based on manual production records, and the number of workers involved was determined at that time. The maximum production level was selected based on the historical production data from the company, representing the highest recorded output. Throughput was calculated according to Equation (1) [36]:(1)Throughput=(GQ+RQ)/AOET
where *GQ* is good quantity, *RQ* is rework quantity, and *AOET* is the actual order execution time (hour). The throughput of the robotic automation system was also calculated based on the quantity produced within the same execution time.

Utilization rate is defined as the ratio of each machine, robot, and worker’s actual work time (*AWT*) to the planned operation time (*POT*). This indicator helps identify worker overload in manual processes. In automated systems, it evaluates whether the robot operates continuously without interruptions caused by bottleneck operations. The utilization rate is determined according to the following equation:(2)Utilization Rate=AWT/POT×100

For both the existing process and the robotic automation system, the Plant Simulation program was configured with common settings, including Availability and the Shift calendar. Availability was set at 95%, representing the percentage of time the equipment was capable of performing the assigned tasks. In the food industry, maintaining high reliability was considered crucial due to the need for very low probabilities of operational issues. The mean time to repair (MTTR) for the robot was set at 15 min, which was defined as the minimum time required to restore the equipment to its initial operating state after a shutdown. The Shift calendar was configured based on a 5-day, 40 h workweek, meaning that the simulation results reflected the performance of the process during an 8 h maximum daily operation.

### 2.5. Model Creation Using Simulation: Robotic Automation System

The movement of the robot and the overall system flow were modeled in Tecnomatix Plant Simulation(V2201), as shown in Figure 4. Using the SimTalk programming language in the method function, commands were configured to use conveyor sensors. This configuration ensured that the tray supports continued moving along the conveyor, while trays designated for product loading stopped precisely at the specified position. When the trays reached the robot’s position, the robot performed the assigned tasks according to the process simulation settings. Once the products were fully loaded onto the tray, it was discharged, and a new empty tray was automatically supplied according to the predefined conditions.

## 3. Results

The analyses focused on the effectiveness of automation compared to traditional manual processes, highlighting key performance indicators such as throughput and utilization. The results are divided into two case studies, each representing different production scales and environments.

### 3.1. Case Study A: Large-Scale Production

#### 3.1.1. Three-dimensional Reconstruction of Food Manufacturing Process

To analyze the utilization rate of workers and the required manpower according to the production volume in Case A, the existing systems and equipment used in the current site were reconstructed in 3D. An automated system incorporating a robotic arm along with an automatic tray separation and supply conveyor system was implemented to evaluate the effectiveness of the proposed solution as shown in Figure 5. The aim was to improve the same area where products are supplied for production. In the existing process, workers are deployed as follows: one worker arranges the supplied products on the turntable, and two workers load products onto the empty trays. Once a tray is fully loaded, the workers stack the empty trays, continuing until a total of 10 layers is reached. Each tray accommodates 24 products, arranged in four rows of six items each. The proposed improvement involves applying a robotic automation system that enables a single worker to handle the entire process.

#### 3.1.2. Model Creation in Process Simulation

In the large-scale production setting of Case A, the process was improved to enhance efficiency. Once the worker aligns the incoming products with a guide and presses a button, the robot picks and places the products in the designated positions. Referring to the existing manual process where workers align products in four rows, the robot’s path was set to follow this alignment, as shown in Figure 6a. To analyze whether the robot’s placement was appropriate, the Smart Place function was utilized. This analysis considered factors such as the robot’s working radius and the arrangement of the automated equipment. Figure 6b shows that the robot is positioned in the optimal blue area within the configured system. During the operation, the interaction between the workers, the robot, and the system was assessed to check for potential collisions. Figure 6c confirms that no collisions occurred, even when the process was repeated ten times, corresponding to loading ten trays onto a single trolley.

#### 3.1.3. Creation in Tecnomatix Plant Simulation

Simulation was utilized to identify issues in the existing manual process performed by workers. As shown in Figure 7, the logic of workers’ (indicated by circles 1, 2, and 3) movements and process execution in the current site was modeled. The equipment was integrated into the simulation to accurately visualize the existing work environment. The simulation results revealed a bottleneck at the turntable waiting process during the product loading operation on trays. It was observed that three workers were producing approximately 24,073 units per day, with an estimated throughput of 3133 units per hour.

The utilization rates of the workers were analyzed separately. The worker responsible for arranging products on the turntable and loading them onto trays had a utilization rate of 79.43%. The worker who transfers products from the turntable to the trays had a higher utilization rate of 94.81%. Meanwhile, the worker who both loads products onto trays and stacks the empty trays had a utilization rate of 60.62%. These results are illustrated in Figure 8.

The results of the overall simulation model with the applied robot are shown in Figure 9. When empty trays are supplied, the automated tray separation system delivers them one by one to the robot’s position. Once the empty trays reach the process, both the worker and the robot carry out their respective tasks. Based on the process speed derived from the simulation, it was analyzed that approximately 4110 products could be processed within the same time frame.

During the production process with the applied robotic automation system, the utilization rates of the worker and the robot were analyzed to be 17.20% and 87.22%, respectively, as shown in Figure 10. This result indicates that the worker’s workload is significantly reduced, while the robot can continue production without interruption. It was also analyzed that the process could be performed with just one worker instead of the three previously required.

To validate the simulation results, the robotic automation system was actually implemented and applied in the field, as shown in Figure 11. The operation commenced with the automated delivery of empty trays and products, replicating the procedure outlined in the simulation. The worker arranged the products in sets of six on the guide and pressed a button, allowing the robot to load the products onto the trays. A sensor was used to ensure that the robot did not operate if no tray was detected. Once the loading was complete, the tray was moved to the automatic stacking system. A new empty tray was then supplied, allowing the robotic automation process to continue smoothly. As a result of performing the process over the same period, the robot’s throughput aligned with the simulation results, processing 4110 units.

The existing manual process, where workers dismantle and supply heavy trays and repeatedly load products, was improved by implementing a robotic automation system. By utilizing simulation, potential errors during installation and setup were minimized before applying the system to the actual site. This approach proved to be highly effective in assessing the feasibility and benefits in advance.

### 3.2. Case Study B: Order-Based Production

#### 3.2.1. Three-dimensional Reconstruction of Food Manufacturing Process

In Case B, the 3D model was created to analyze the utilization rates and manpower of workers during peak production periods in an order-based production case. The existing systems and equipment used at the current site were reconstructed in 3D. As shown in Figure 12, this reconstruction was performed to evaluate the effectiveness of implementing a robotic automation system in the same environment. In the current process, one worker loads products onto trays while another supplies empty trays. Once the trays are fully loaded, the worker responsible for tray separation moves the cart to the retort sterilization process, completing the operation. The process continues until a total of 10 trays are loaded onto a single cart, with each tray holding 12 products arranged in two rows of six. The proposed improvement involves applying a robotic automation system. With this system, the worker’s task is simplified to just supplying empty trays and removing fully loaded trays, making the process more efficient.

#### 3.2.2. Model Creation in Process Simulation

In Case B, a process simulation was created where the robot automatically loads products into pre-determined tray positions. This setup assumes that six products are consistently supplied as part of an order-based production system. The robot’s path was configured, as shown in Figure 13a, with numbers 1 and 2 indicating the sequential steps of the robot’s movement. Given the limited workspace and the need for fully automated product loading without workers, the robot’s speed was intentionally set slowly. The suitability of the robot’s placement was analyzed, as depicted in Figure 13b, confirming that it was positioned in the optimal blue area, while the red areas indicate unsuitable positions. The simulation also investigated potential collisions and interferences between the robot, the walls, and the automated system components. Figure 13c illustrates that no collisions occurred during the 19 iterations of the process.

#### 3.2.3. Creation in Tecnomatix Process Simulation

To analyze the utilization rates of workers during production at the existing site, a 3D simulation of the actual work environment was created, as shown in Figure 13. The workers’ movement paths, process execution, and tray transport were modeled in a logic-based system, as illustrated in Figure 14a. The existing equipment was then integrated into the 3D environment to visualize the operation, as depicted in Figure 14b.

The simulation included the manual process, where a worker produces approximately 648 units per hour, to analyze the worker’s utilization rate. The utilization rate of the worker responsible for loading products onto trays was analyzed to be 99.47%, as shown in Figure 15. The high utilization rate shows that the worker is overburdened and constantly performing tasks without any breaks. This emphasizes the urgent need for improvements, such as introducing a robotic automation system. In contrast, the worker responsible for retrieving the cart with empty trays, waiting, supplying trays, and transporting the loaded cart had a utilization rate of just 6.11%. In the actual site, complex tasks and movement paths often lead to confusion. To address this, the simulation simplified these elements by separately defining work activities for analysis. This approach implies an inherent imbalance in the workers’ tasks.

The results of applying the robotic automation system to improve the manual process in Case B are shown in Figure 16. Considering the limited workspace, tray transport conveyors were arranged to circulate and return empty trays to the worker’s position. The robot was positioned at the product supply area to automatically load the products onto the trays. The plant simulation results indicated that the robot could process approximately 726 products within the same time frame.

As shown in Figure 17, the robot’s utilization rate was 99.41%, confirming that the robot can continuously handle production without interruption. The utilization rate of the worker responsible for supplying and retrieving the carts was approximately 2.62%, indicating that the entire process can be managed by a single worker.

To validate the results of the robotic automation system derived from the simulation, it was implemented in the actual site as shown in Figure 18. The process began when the worker supplied the trolley loaded with empty trays, and production started simultaneously as the products were produced. The system was designed so that once six products were automatically aligned and supplied, the robot would start its operation. The robot-initiated movement after receiving confirmation signals that both the products and empty trays had arrived. In practice, the empty trays were supplied and fully loaded with products, and the robot automation system achieved a throughput of approximately 723 units per hour.

### 3.3. Comparison of Each Cases

The applications of the robotic automation system in both Case A and Case B demonstrated clear improvements compared to the existing manual processes. The results indicate that product throughput and handling capacity increased over the same time period, while the utilization rate and number of workers required for the process decreased significantly (Table 1).

In Case A, the throughput increased by 31.2% after the introduction of the robotic automation system, with the same improvement seen in both the simulation and actual application. In Case B, the throughput increased by approximately 12.0%, and although there was a minor discrepancy between the simulation and actual results (a difference of only three products), this was considered negligible. The slight variation can be attributed to inconsistencies in the process steps before or after the analyzed process, which are common in real production environments. Both case studies highlight the effectiveness of the robotic automation system in not only enhancing productivity but also in reducing the physical demands placed on workers, allowing for streamlined and more efficient operations.

## 4. Discussion

This study explored the potential of applying robotic automation systems in food manufacturing processes. While previous studies have primarily focused on general manufacturing settings, this research specifically addressed challenges in the HMR sector. Simulations were used to assess the performance and feasibility of transitioning from manual to automated systems. The primary objective was to identify issues such as worker overload and bottlenecks in existing manual processes, and to evaluate how these could be improved through automation.

The simulations analyzed the utilization rates of workers and identified bottlenecks during process execution. It confirmed that implementing a robotic automation system can effectively address these inefficiencies. These results aligned with previous studies demonstrating that robotics showed significant potential to enhance production efficiency in manufacturing [16,17,18,19,20]. The comparison between the worker utilization rates in the manual system and the robot’s performance after automation showed significant improvements in process efficiency. Both Case A and Case B demonstrated the system’s applicability across various production environments by applying a similar robotic automation system under different conditions.

The validation of the simulation results through actual site application is a significant contribution of this study, as it confirms the practical utility of simulation models in predicting production improvements. This validation showed that the simulated outcomes closely matched the actual results, affirming the reliability of the simulation models. The similar results between the simulation and real application show that simulations are useful tools for planning and improving production processes.

To determine the effectiveness of the robotic automation system, KPIs such as throughput and utilization rate were selected. These KPIs are critical in evaluating production-related factors and were used to compare the performance of manual and automated systems. Additionally, the number of workers involved was also analyzed to assess the system’s capability to replace human labor. The findings indicate that robotic automation can reduce the physical strain on workers while maintaining or even increasing production output.

Despite the promising results, several limitations must be acknowledged from this study. The simulations were based on specific case studies and may not fully represent all possible production environments. This study focused exclusively on pouch-type HMR products with specific dimensions of 150 mm (Length) × 256 mm (Width). Therefore, the findings could show different results when applied to different types of HMR products or other manufacturing processes. Case A allows for intermediate control by workers during production. In contrast, Case B operates under the assumption of a fixed production rate of six units per batch, which may not accurately reflect other production cases. Additionally, the study did not address the cost implications of implementing robotic systems, which could affect the feasibility for SMEs. As only unit processes were selected for analysis, future research should consider a broader range of production environments. It is necessary to gradually expand the scope by considering preceding and subsequent processes, allowing for an eventual full-scale factory simulation that supports digital transformation. It should also conduct detailed cost–benefit analyses and explore the integration of robotic systems with human labor to better understand their practical applications.

This research demonstrates the usefulness of simulation models in designing and testing production systems. This study highlights the potential for robotic automation to enhance production efficiency and worker safety in the food industry. For SMEs, this study provides practical insights into overcoming operational challenges associated with manual processes. It suggests that integrating automation can be significantly helpful in productivity while also reducing labor costs. The case studies show that robotic automation systems can be effectively integrated into existing production environments, offering significant productivity gains and operational consistency. Furthermore, the study illustrates that similar automation strategies can be applied across different production settings, making it a versatile solution for the food manufacturing industry.

Further research could focus on exploring more diverse production environments and investigating different types of robotic system configurations, including delta robots commonly used in food packaging, to determine their effectiveness across various stages of food processing. It could also analyze the cost-effectiveness of implementing such systems in SMEs. Additionally, investigating hybrid systems that combine both human and robotic labor could provide further insights into optimizing production efficiency in various contexts.

## 5. Conclusions

In recent food manufacturing, many companies are increasingly adopting large-scale automation systems to ensure consistent, safe, and high-volume production. The integration of robotics and various sensors is driving the transition toward digitalization and smart factories. This study evaluated the effectiveness of applying a robotic automation system using simulation-based methods in two different manually operated production environments. The results showed that the simulated outcomes were similar to the real applications.

Bottlenecks and worker overloads in existing manual processes were identified through simulation. The application of the robotic automation system was shown to significantly reduce worker utilization rates while increasing overall production output. The study confirmed that the simulation results reliably predicted the effectiveness of the robotic automation system when applied in actual production settings. This suggests that the proposed approach could be applied to various food manufacturing environments to assess and enhance automation strategies. Furthermore, effectively implementing and utilizing the 3D simulation models from this study in real applications will require careful consideration of various factors. These factors include actual production times from preceding and succeeding processes. By integrating data collection, AI algorithms, and comprehensive plant simulations like those conducted in this research, it will be possible to develop systems that predict full-process production. These systems can also optimize energy use across different stages. This also could lead to the development of more efficient and adaptive automation systems optimized for diverse food production environments.

## Figures and Tables

**Figure 1 foods-13-03121-f001:**
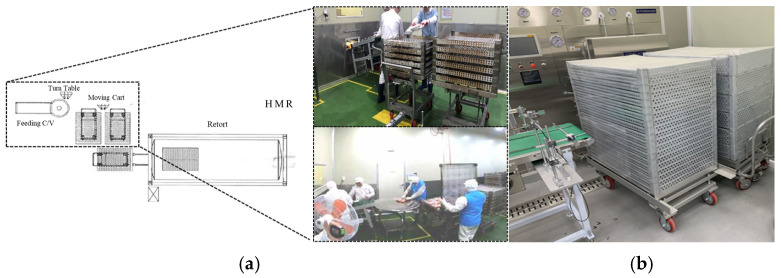
Manual process of loading pouch products onto retort trays in each case: (**a**) Large-scale production, (**b**) Order-based production.

**Figure 2 foods-13-03121-f002:**
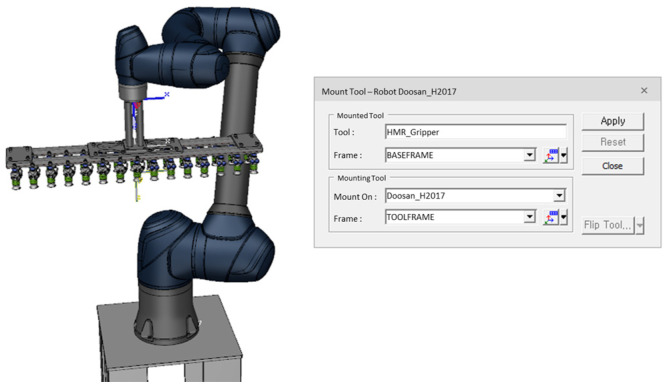
Mount vacuum gripping system of the end-effector to robot and defined as Tool Center Point (TCP) in Tecnomatix Process Simulate.

**Figure 3 foods-13-03121-f003:**
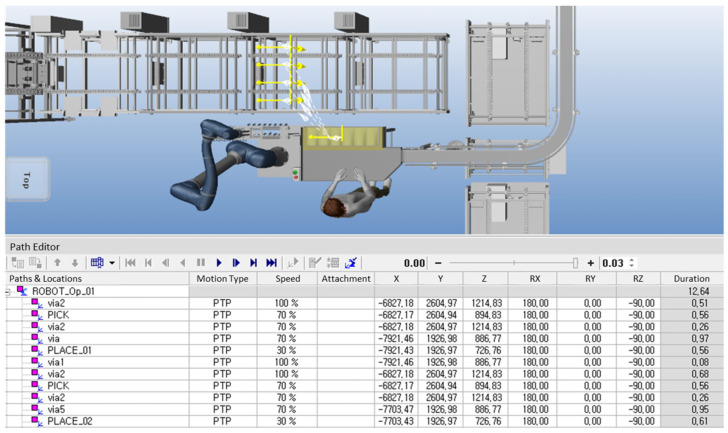
Control robot speed and trajectory of robot in Tecnomatix Process Simulation.

**Figure 4 foods-13-03121-f004:**
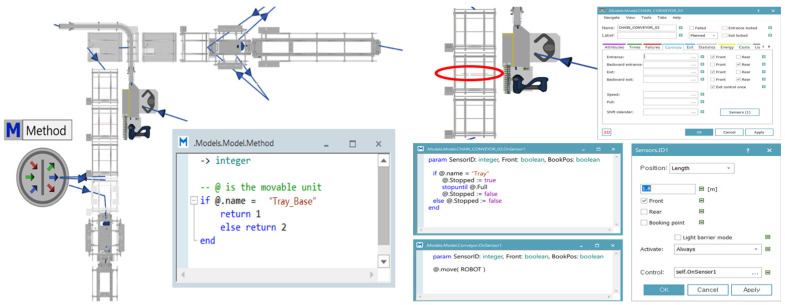
Proposed simulation model with the applied methods.

**Figure 5 foods-13-03121-f005:**
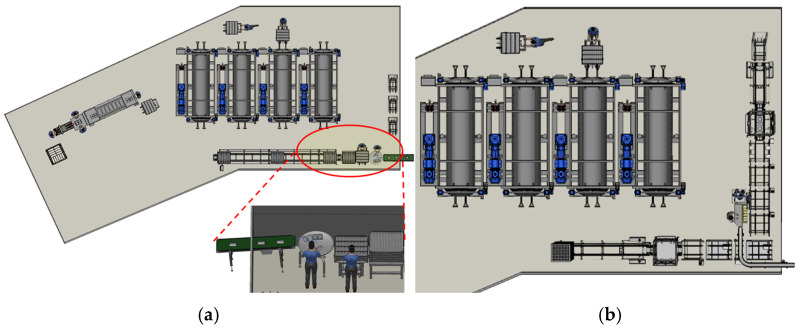
Simulation model of case A: (**a**) Existing site of the food manufacturing process, (**b**) Robot application system.

**Figure 6 foods-13-03121-f006:**
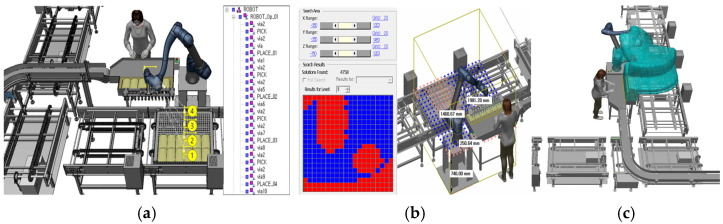
Planning the robot’s path for process execution and evaluating the suitability of the robot’s placement in Case A.

**Figure 7 foods-13-03121-f007:**
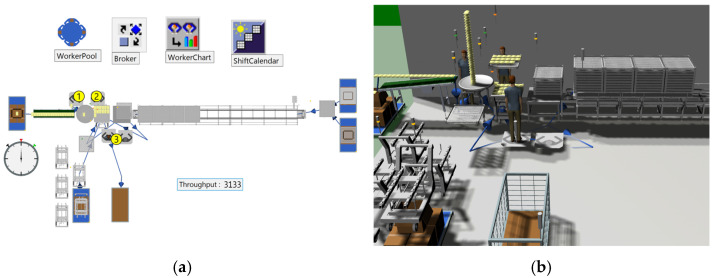
Analyzing throughput and utilization rate using plant simulation in Case A: (**a**) Logic of the manual process, (**b**) 3D simulation of the manual process.

**Figure 8 foods-13-03121-f008:**
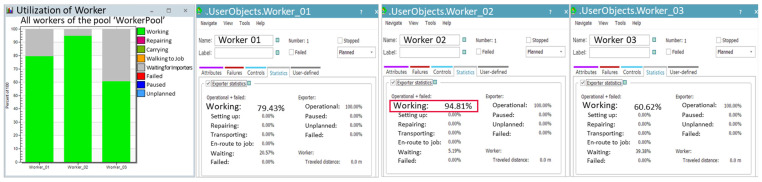
Utilization rates of each worker in the manual process for Case A.

**Figure 9 foods-13-03121-f009:**
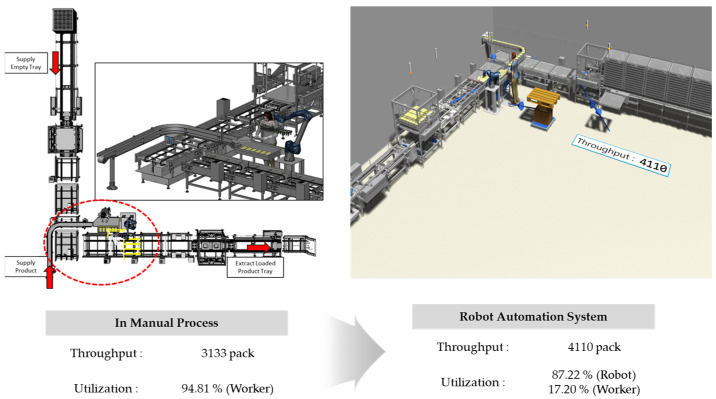
Analysis of throughput and utilization rate of the robot automation system using plant simulation in Case A.

**Figure 10 foods-13-03121-f010:**
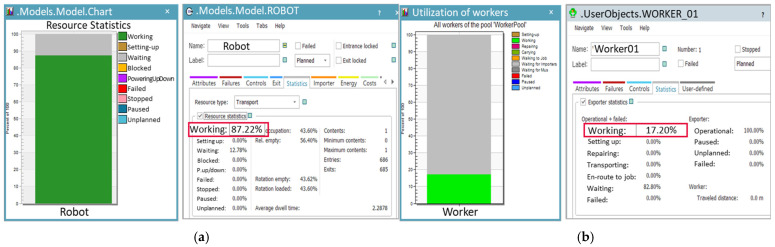
Utilization rates of the robot (**a**) and worker (**b**) in the robot automation process for Case A.

**Figure 11 foods-13-03121-f011:**
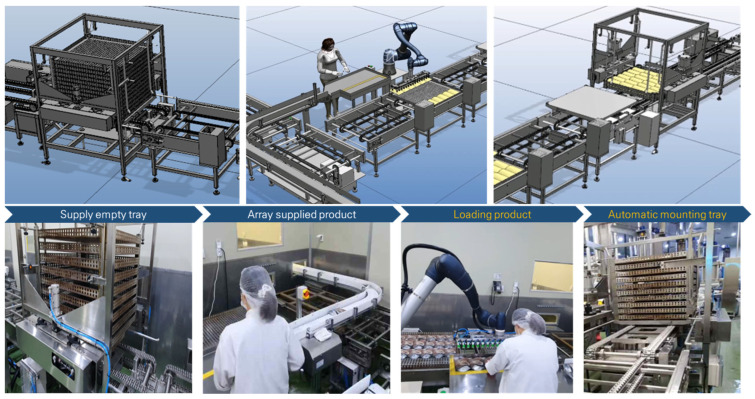
Application of the robot automation system process, improved by simulation, to the actual site in Case A.

**Figure 12 foods-13-03121-f012:**
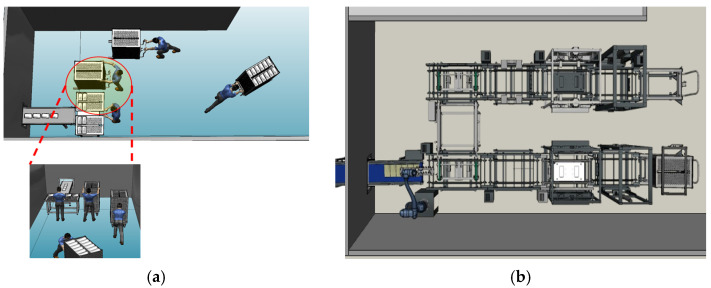
Simulation model of case B: (**a**) Existing site of the food manufacturing process, (**b**) Application of robot automation system.

**Figure 13 foods-13-03121-f013:**
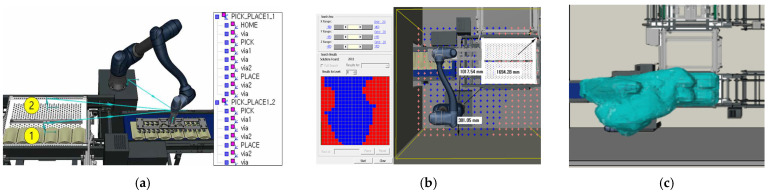
Planning the robot’s path for process execution and evaluating the suitability of the robot’s placement in Case B.

**Figure 14 foods-13-03121-f014:**
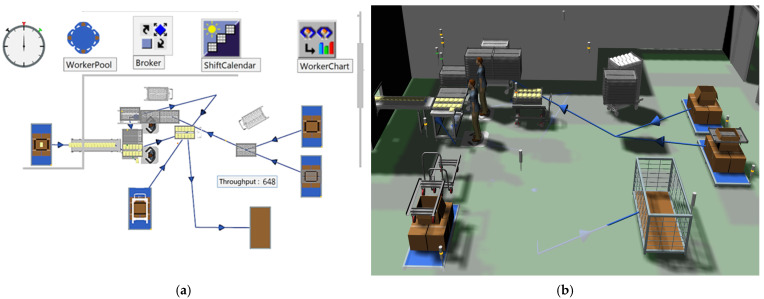
Analyzing throughput and utilization rate using plant simulation in Case B: (**a**) Logic of the manual process, (**b**) 3D simulation of the manual process.

**Figure 15 foods-13-03121-f015:**
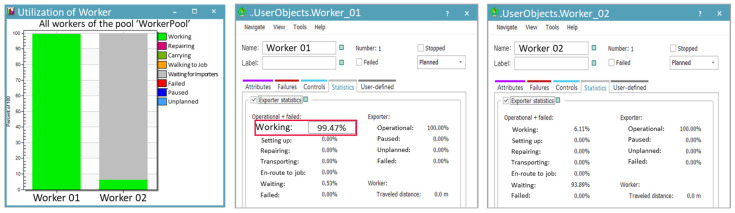
Utilization rates of each worker in the manual process for Case B.

**Figure 16 foods-13-03121-f016:**
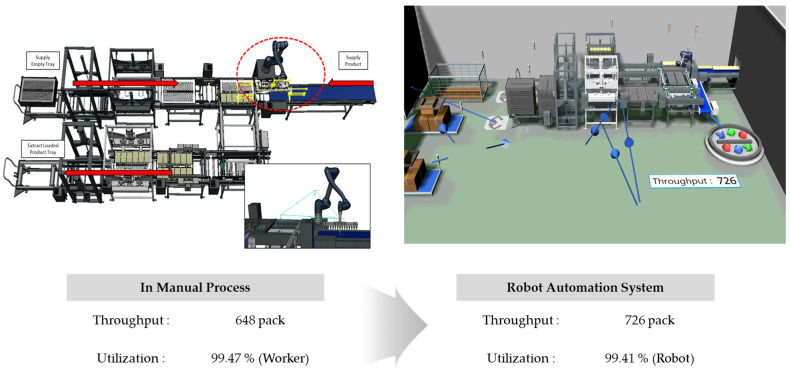
Analysis of throughput and utilization rate of the robot automation system using plant simulation in Case B.

**Figure 17 foods-13-03121-f017:**
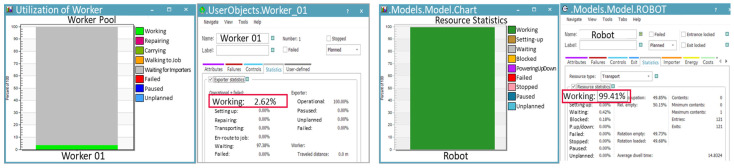
Utilization rates of the worker and robot in the robot automation process for Case B.

**Figure 18 foods-13-03121-f018:**
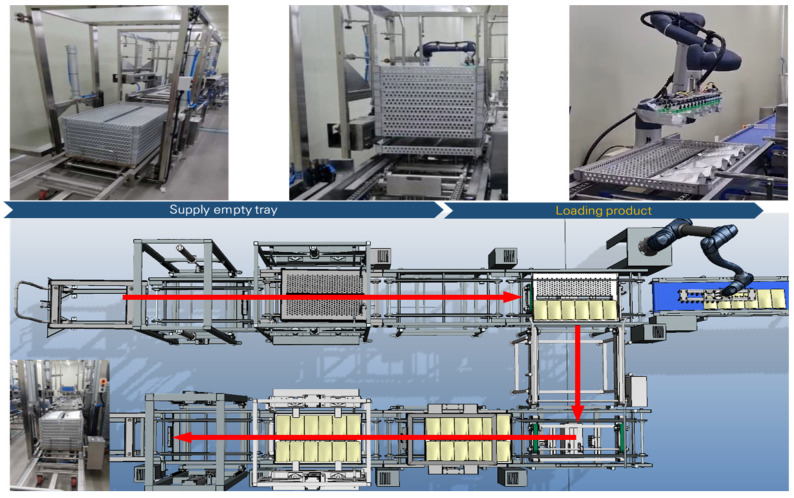
Application of the robot automation system process, improved by simulation, to the actual site in Case B.

**Table 1 foods-13-03121-t001:** Comparison of effectiveness in throughput, utilization rate, and employee involvement between simulation and actual application in each case.

Category		Simulation			Real Application	
		As-Is (Manual)	To-Be (Robot Automation)	Improvement Rate (%)	To-Be	Improvement Rate (%)
Case A	Throughput (pack)	3133	4110	31.2	4110	31.2
	Utilization Rate (%)	94.81 (Worker)	87.22 (Robot)	-		
	Involved Employee (worker)	3	1	66.7	1	66.7
Case B	Throughput (pack)	648	726	12	723	11.6
	Utilization Rate (%)	99.47 (Worker)	99.41 (Robot)	-		
	Involved Employee (worker)	2	1	50	1	50

## Data Availability

The original contributions presented in the study are included in the article, further inquiries can be directed to the corresponding authors.

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
