# Peer review of "A Simulation-Based Approach for Evaluating the Effectiveness of Robotic Automation Systems in HMR Product Loading"

_foods, 2024, doi:10.3390/foods13193121_

Round 1
Reviewer 1 Report
Comments and Suggestions for Authors
This article presents an approach for using simulation in development of applications in the food industry. The manuscript is interesting, but some issues should be dealt with:
What is specific to food industry? Are there any standards?
Why don’t you use DT solutions for simulation-based approaches?
Types of robots: what about delta-robots which are frequently used in food processing (e.g.: packaging)?
It is not clear how you validate trajectories – avoid obstacles, adjust speed, handler/gripper design, compute throughput, cycle time. Simulation in robotics is more than diagram-blocks and flowcharts.
In my opinion there is no gap in the studies. The authors should check the concept of model-based digital twin for application design and simulation.
Section 2.3 presents a specific robot. The work presented here should be generic.
Rows 163-166 are ambiguous. The structure/shape of the tool in is important.
Equation 1 is the definition of throughput, what is particular in your case scenario? Equation 2 has the same particularities as Equation 1.
What is the application for developing the 3D model? Is it Tecnomatix Plant Simulation?
Fig.8 is too small. Readers cannot read it. Same for 10.
Fig.13c is the working envelope. Besides the conveyor there is no obstacle (fence, other resource). Please clarify how you detect collisions (in RobotStudio there are sets of objects that when they interact events are risen). How is it done in your application?
Fig.14a – very small
Fig.15 is clear, but small
Fig.17 small
Table 1 is not clear, simulation is as the real application?
Author Response
The authors appreciate the reviewer's constructive comments and suggestion.

Reviewer 2 Report
Comments and Suggestions for Authors
The paper provides a comprehensive evaluation of robotic automation systems in Home Meal Replacement (HMR) product loading through simulation. It offers valuable insights into the operational efficiency and potential benefits that such systems can bring to small and medium-sized enterprises (SMEs). The use of discrete event simulation (DES) to analyze two different production scenarios (large-scale and order-based) is methodologically sound and contributes positively to the literature on food production automation.It was an honour to review your article and your research direction is very interesting, but I have some suggestions about your article:
Introduction:
Lines 31-40: Please identify the corresponding context and examples.
Discussion
1. While the article briefly mentions the limitations of the study, the specific impact of these limitations on the interpretation of the results could be discussed more fully. For example, the specific production environment and HMR product type on which the simulations were based may have limited the general applicability of the results.
2. It is recommended that the discussion section elaborate on possible directions for future research, such as exploring different types of robotic system configurations or the potential for applying the technology in other food production environments.
3. Some of the graphs and visual presentations could be further optimised to show relationships and comparisons between Key Performance Indicators (KPIs) more clearly. The use of more graphical analyses may help readers to better understand the data.
4. The article could provide more specific recommendations and guidance on the challenges and coping strategies that SMEs may encounter when implementing such systems, which would increase the value of the study for practical application.
Comments on the Quality of English Language
no
Author Response

(The authors gave the same response as above.)

Reviewer 3 Report
Comments and Suggestions for Authors
The study aims to evaluate the impact of robotic automation on improving production efficiency in the Home Meal Replacement (HMR) industry using a simulation-based approach. The subject of the paper is interesting and in line with the aims and scope of the Journal. The paper is well-structured and well-written. It provides interesting and valuable results. However, the novelty and contributions of the study are not highlighted well enough. Some parts of the paper should also be improved, in particular introduction and discussion. More detailed comments are provided below.
1. The authors should highlight the contributions and novelty of the study in the Abstract.
2. The introduction is not written well. It is more of a literature review than an introduction. The authors did not place the study in a broad context nor highlight why it is important to investigate this subject. They should have defined the purpose of the work and its significance, including specific hypotheses being tested. The authors did not point out the main aim of the work nor highlight the main conclusions and scientific contributions of the paper. At the end of the introduction, the authors should have provided a brief overview of the following sections. The paper lacks a proper introduction since most of the material from the current introduction should be moved to a separate section under the heading “Literature review”.
3. The authors should provide additional information about the methodology. It is clear that they used simulation software, but what kind of algorithms does this software use? A literature review on this topic is also missing.
4. The discussion is weak. The discussion should discuss and interpret their research from the perspective of the previous studies in terms of the results and the methodology.
5. The authors should explain the theoretical and practical (managerial) implications of their study.
6. Some technical issues should be addressed:
a) There should be at least a couple of sentences between the headings of different levels (e.g. between section 2 and sub-section 2.1.).
b) Some figure captions are too extensive (e.g. for Figures 6, 13, etc). If you need to describe the figure additionally, do it in the main text. Figure and table captions should be short and informative.
c) The text in some figures is hardly visible (e.g. in Figures 8, 10, 15, etc).
d) Table 1 is not formatted according to the Instructions for authors (provided template).
Comments on the Quality of English Language
The English language is acceptable. Only minor issues are noticed.
Author Response

(The authors gave the same response as above.)

Round 2
Reviewer 1 Report
Comments and Suggestions for Authors
the article has been improved according to the observations.
Reviewer 2 Report
Comments and Suggestions for Authors
The manuscript has been carefully revised and can be accepted and published.